# The Effects of Light Spectrum and Intensity, Seeding Density, and Fertilization on Biomass, Morphology, and Resource Use Efficiency in Three Species of *Brassicaceae* Microgreens

**DOI:** 10.3390/plants13010124

**Published:** 2024-01-01

**Authors:** Reed John Cowden, Bo Markussen, Bhim Bahadur Ghaley, Christian Bugge Henriksen

**Affiliations:** 1Department of Plant and Environmental Sciences, University of Copenhagen, Højbakkegård Alle 30, 2630 Taastrup, Denmark; bbg@plen.ku.dk (B.B.G.); cbh@plen.ku.dk (C.B.H.); 2Department of Mathematical Sciences, University of Copenhagen, Universitetsparken 5, 2100 København Ø, Denmark; bomar@math.ku.dk

**Keywords:** microgreens, LED, light recipes, fertilization, seeding density, resource use

## Abstract

Light is a critical component of indoor plant cultivation, as different wavelengths can influence both the physiology and morphology of plants. Furthermore, fertilization and seeding density can also potentially interact with the light recipe to affect production outcomes. However, maximizing production is an ongoing research topic, and it is often divested from resource use efficiencies. In this study, three species of microgreens—kohlrabi; mustard; and radish—were grown under five light recipes; with and without fertilizer; and at two seeding densities. We found that the different light recipes had significant effects on biomass accumulation. More specifically, we found that Far-Red light was significantly positively associated with biomass accumulation, as well as improvements in height, leaf area, and leaf weight. We also found a less strong but positive correlation with increasing amounts of Green light and biomass. Red light was negatively associated with biomass accumulation, and Blue light showed a concave downward response. We found that fertilizer improved biomass by a factor of 1.60 across species and that using a high seeding density was 37% more spatially productive. Overall, we found that it was primarily the main effects that explained microgreen production variation, and there were very few instances of significant interactions between light recipe, fertilization, and seeding density. To contextualize the cost of producing these microgreens, we also measured resource use efficiencies and found that the cheaper 24-volt LEDs at a high seeding density with fertilizer were the most efficient production environment for biomass. Therefore, this study has shown that, even with a short growing period of only four days, there was a significant influence of light recipe, fertilization, and seeding density that can change morphology, biomass accumulation, and resource input costs.

## 1. Introduction

Current agricultural practices constitute at least 17% of total global GHG (CO_2_-equivalent) emissions [1], while also threatening water and soil resources via eutrophication, acidification of water, and increasing emissions of N_2_O [2,3]. At the same time, there is a growing concern for the future availability and consequences of overuse of phosphorous and potassium in fertilizers [4,5,6]. Furthermore, changing regional patterns of rainfall, increasing frequency of disastrous weather events, human conflicts, and projected increases in temperature threaten food systems, which lack resilience in their monocropping format, which in turn decreases food security [2,7,8,9,10,11]. Therefore, status quo practices are not sustainable. A food production alternative gaining increasing attention is Plant Factories with Artificial Lighting (PFALs), which are indoor, insulated, environmentally controlled production systems with artificial lighting that are especially useful for producing leafy greens or valuable specialty crops such as microgreens. Microgreens are nutritious specialty crops that have a short growing period (around 7–21 days depending on species or desired output), high Seeding Density (SD), and short height, making them optimal for tight, layered growing spaces that can be grown year-round with artificial light sources [12]. They are harvested between the phenological stages of sprouting and baby greens, usually at or around the appearance of the first true leaves [13,14,15]. In general, they can be cultivated from the seeds of almost any vegetable, herb, or grain species [16], but there are around 80–100 commonly cultivated species of microgreens in use today [17], with *Amaranthaceae*, *Apiaceae*, *Asteraceae*, and *Brassicaceae* families being the most common [13]. Although microgreens are widely used for garnishing in the culinary industry [18], they are increasingly gaining attention as potential ‘functional foods’ that can help fortify human nutritional needs because of their elevated concentrations of vitamins, minerals, and bioactive compounds compared to mature plants [19,20,21,22,23]. Microgreens can also have lower levels of nitrate [24], which is regulated as a carcinogenic substance [25]. Furthermore, microgreens can also be produced in areas where traditional vegetable cultivation is difficult, such as arid or alpine environments [26,27]. Growing leafy greens like microgreens via PFALs can also substantially reduce the input requirements of water and nutrients since these resources can be recycled within the system without major losses, while also allowing spatial use efficiency to increase [28]. These benefits show that microgreens should be evaluated as a method for efficiently delivering healthy plant biomass and nutritional compounds to fortify the growing global human population’s needs.

One of the most important environmental factors when producing microgreens is light, as it can augment desired production outcomes. Light is an essential part of PFAL systems for two primary reasons: firstly, it supplies the required energy input to drive photosynthesis, and secondly, it cues the plant via photomorphogenic activation or deactivation of pigments and photoreceptors; for example, a low Red:Far-Red ratio can induce increases in plant height [29,30,31,32]. Therefore, the specific light spectral components of ‘Light Recipes’ can induce physiological modifications and shift developmental pathways, which necessitates careful consideration of their design [33]. Although light is known as a critical component of plant production processes, much research has focused on dichromatic Light Recipes that are primarily composed of Red and Blue light components. This is due in part to early foundational research that detailed plant absorption and the quantum efficiency of intercepted wavelengths of light and showed that plant absorption was lower for light within the range of the Green wavelength (500–600 nm) than the Red and Blue spectral bands (600–700 nm and 400–500 nm, respectively) [34,35,36]. However, this high absorbance firstly does not necessarily translate into photosynthetic use, as photosystems can become saturated, with excess light being shed as nonphotochemical quenching (i.e., heat); secondly, it does not consider the synergistic effects of other bands of light, both within and outside the PAR range, also known as the ‘Emerson Effect’ [37,38,39]. Given this research on apparently low-absorptance, or low-utility, spectral bands such as Green light or Far-Red light, which are listed as outside the range of Photosynthetically Active Radiation (400–700 nm), fewer studies have focused on the addition of these light components in spectral recipes [37,40]. However, more recent contemporary research has illustrated the potential advantages of including these spectral bands [41,42,43,44,45,46,47], which necessitates further research in this regard, especially for microgreens, where fewer studies have been conducted [47]. One increasingly important topic is using Far-Red light to manipulate morphology and photosynthetic efficiency [48,49,50]. This not only saves energy in sole-source lighting scenarios but can also potentially cause less photooxidative stress to plants, as Far-Red (700–800 nm) is a longer wavelength with less energy per photon compared to shorter wavelengths.

However, research on the productivity or morphological advantages of plants due to incident light wavelengths has often come without reporting the associated costs of production to achieve these inputs. When investigating sustainable food production practices for the future, it is important to optimize yield but also to consider resource use efficiency (RUE). Despite the advantages of PFALs reviewed by Orsini et al. [28], there are several drawbacks, such as the high cost of setup [51,52], high material intensity [53], and lower energy efficiency compared to outdoor food production activities [54]. Considering light is the major energy input required for PFALs, it is important to determine how different light types and corresponding intensities and spectra affect both microgreen production and corresponding RUE. As PFALs could be an essential tool to buffer against food insecurity in the future [55], understanding the RUEs of different production environments can offer much novel value addition regarding microgreens production.

Therefore, in this study, our first objective was to measure kohlrabi, mustard, and radish microgreen biomass and morphology outcomes as a function of five different Light Recipes grown at two seeding densities with and without fertilization. Our second objective was to contextualize these results by identifying RUE pathways. We hypothesized that, by modulating the incoming light at different treatment levels, we could change the production, morphology, and RUE of these microgreens. Specifically, by including higher components of Far-Red light, we expected improvements in height and leaf characteristics that could result in greater biomass accumulation via various improvement pathways, from increased light interception to potential photosystem balancing; these improvements were contextualized by the inclusion of RUE results to also discuss the tradeoffs of these production methods. Furthermore, we expected that the PAR-dichromatic Light Recipe would perform less well than the other Light Recipes due to the advantages of the Emerson Enhancement Effect in the other Light Recipes; this is also expected to have an effect due to the highly dense, layered canopy of microgreen production systems. Finally, we also wanted to quantify the impacts of Seeding Density and Fertilization. Quantifying the biomass accumulation outcomes of microgreens subjected to these factors can help identify tradeoffs and pathways for optimizing outcomes that can inform food-secure production systems.

## 2. Materials and Methods

### 2.1. Experimental Context

This experiment was conducted in 2022 at the University of Copenhagen, Denmark, in a programmable climate chamber (Conviron CMP5090; Winnipeg, MB, Canada). Within the climate chamber, we constructed a six-tiered growing system, supplied by a commercial microgreen producer (Instagreen.eu, Barcelona, Spain). The grow system measured 112 cm (L), 120 cm (W), and 217.5 cm (H) and could hold up to 12 trays on six layers (2 trays per layer), with a total usable growing space of 7.26 m^2^. Each growing tray has dimensions of 110 cm (L), 55 cm (W), and 5 cm (H). Each tray can fit forty cups, in which the plants are grown, with each cup measuring 14.5 cm (L), 10 cm (W), and 5 cm (H). In addition to the externally located climate control software, we also utilized two ‘Paladin pro4 Bluetooth’ timers (Hugo-mueller.de, Villingen-Schwenningen, Germany) to program schedules for the water pump and the ‘Fullwat DOMOX line high-efficiency 24-Volt’ LEDs (24VHELED; Fullwat, Bilbao, Spain) using the ‘Save’n carry’ application for Android (Hugo-mueller.de). The second set of lights were OSRAM Phytofy Research Lights (Osram, Beverley, MA, USA), which were programmed using the ‘PHYTOFY’ software v1.0. We set our grow system to be watered two times a day for 15 min, once at the beginning of the 16 h photoperiod, and once again 8 h later.

Our experimental design used the upper three layers as growing spaces, with the top-most layer having three OSRAM lights that were 41 cm from the emitters to the surface of the trays; each light was treated as a block. The second and third layers each used four 1 m-long 24VHELEDs equidistant from one another, which were 32 cm from the emitters to the trays; each of the 3 inter-light spaces was a block. The bottom three layers were used as germination space, where each layer consisted of four standard growing trays, two on the bottom and two inverted on the top, to create a high-humidity dark zone; the trays were slightly off-set so that the water could run through the system as well as regulate the relative humidity (RH). The microgreen seeds were sown in plastic cups on 100% cellulose growing pads according to the sowing densities listed in Appendix A. The substrate was soaked until saturated with distilled water prior to sowing. None of the seeds were pre-soaked. However, after sowing seeds, they were misted for approximately 1 s with distilled water to ensure initial moisture at the superior surface. Our substrate was not compensated with a top-weight over the seeds, which was crucial for allowing the seedlings unimpeded vertical growth for the short dark-zone germination and development time of 4 days. After sowing, each of the 6 cups—2 Seeding Densities (SD) per 3 species—were allocated according to a Complete Random Block Design (CRBD); for a total of 18 cups (3 replicate blocks) per layer per experimental generation. After the 4-day dark-germination period, the plants were put under the lights for 4 days of light treatments and were therefore harvested 8 days after sowing. The first round of experiments was conducted from 12 May to 11 July 2022, while the second round was conducted from 21 August to 14 October 2022.

These experiments used three species of Brassicaceae microgreens: kohlrabi (*Brassica oleracea* convar. Acephala var. gongylodes L. cv. Red Cardinal), mustard (*Sinapis alba* cv. White Candy), and radish (*Brassica oleracea* cv. Daikon Panzer). Standard SD values were derived from standard practice by microgreen producers; the High SD treatment had 50% more seeds in addition to the standard SD. The seeds were sourced from the same supplier as our grow system, Instagreen (Instagreen.eu, Barcelona, Spain). Additionally, suggested Standard seed sowing densities were also provided by Instagreen. The microgreen SD, 100 seed weight, estimated seeds per cup for each sowing density, and germination percentage can be seen in Appendix A, shown as a mean ± the standard error (SE). Overall, this table shows that we achieved very good germination results for radish and mustard at around 99%, while for kohlrabi, the germination rate was around 91.5%. However, we saw good, uniform production of kohlrabi (low SE), and it was not thought to impact the results.

### 2.2. Light Conditions

Within horticultural research, incident light can be defined as the number of photons intercepted by the plant per unit area over time, which is referred to as Photosynthetic Photon Flux Density (PPFD) in units of micromoles per meter squared per second (µmol/m^2^/s). PPFD describes photons within the wavelength range of 400–700 nm, which is the range of Photosynthetically Active Radiation (PAR) [56]. However, recent research has challenged the traditional range of PAR by calling to include Far-Red (around 700–750 nm) in definitions of ‘useful’ light for plants [49,57]. Therefore, because we have included Far-Red in our Light Recipes, we described our light conditions using Photon Flux Density (PFD), which includes all photons within the range of 350 to 800 nm. However, it is also important to consider Irradiance when defining incident light conditions, which is the radiant flux of the received photons for a given unit of area in standard units of Watts per m^2^ (W/m^2^). The ratio of PFD to Irradiance is different for each wavelength of light. That is, photons have different energies at different wavelengths. For instance, 100 µmols/m^2^/s of Red light has less energy (i.e., Irradiance) than 100 µmols/m^2^/s of Blue light; therefore, it also costs more energy to theoretically produce this amount of Blue light; practically, it also depends on the conversion efficiency of the LEDs. This is important to consider, as not all Light Recipes have the same Irradiance, even if they have the same PFD, and vice versa. The link between Irradiance and the number of photons can be seen in Appendix A (in Appendix A), which was calculated from our spectrometric measurements. It shows that the number of photons (µmol) that are produced each second per milliwatt of irradiated energy at each wavelength (350–800 nm) has a linear function with R^2^ = 1.00. Therefore, choosing Light Recipes composed of lower wavelengths can be more energetically expensive, which is an important cost consideration.

The OSRAM lights used in this experiment have a max power draw of 150 Watts per fixture, with dimensions of 66.7 cm (W), 29.9 cm (L), and 4.4 cm (H). They have a total luminaire efficiency of 96.77% with five channels of spectral peaks at 385 nm (UV-A, maximum emittance 50 µmol/m^2^/s), 450 nm (Blue, max 250 µmol/m^2^/s), 521 nm (Green, max at 100 µmol/m^2^/s), 660 nm (Red, max at 250 µmol/m^2^/s), and 730 nm (Far-Red, max at 100 µmol/m^2^/s), each peak of which can be set independently as a percent (0–100%) of its total output. There is also a sixth ‘White’ LED channel, which has a color temperature of 2700 Kelvin—classified as warm light near ‘Sunrise and Sunset’—and a maximum emittance of 250 µmol/m^2^/s. Therefore, the total theoretical maximum PFD for each OSRAM research light is 1000 µmol/m^2^/s. Our second type of light, the 24VHELEDs, have a light color temperature of 4000 Kelvin, near direct sunlight (4800 K), an output of 1380 lumens/m, a power of 12 W/m, a Color Rendering Index ≥ 83, and a PFD of 46 µmol/m^2^/s.

Table 1 details the spectrometric data for our five Light Recipes used in these experiments. Data were derived from a ‘PG100N Spectral PAR Meter’ (UPRtek, Taiwan) with a sensitive wavelength range of 350–800 nm and illuminate accuracy of ±5%. These spectral measurements were taken where the aperture (light sensor) was 10 cm above the growing surface to approximate the average conditions experienced at microgreen canopy height. This was used to measure the Irradiance (W/m^2^), PFD, and PPFD (µmol/m^2^/s), among other spectral properties, of our different Light Recipes. We also calculated the Yield Photon Flux Density (YPFD)—the product of relative quantum efficiency—for each Light Recipe, derived from [34]. Furthermore, we calculated the Phytochrome Photoequilibrium (PPE) according to calculations from [58]. To measure the specific energy use (in Watts) of our Light Recipes, we used a ‘Zimmer LMG610′ (Frankfurt, Germany). For creating a standard Light Recipe, we referenced a ‘Vegetative’ Light Recipe from Valoya (Helsinki, Finland), a horticultural light company, which was replicated as the ‘High Red’ recipe in our experiments.

Appendix A, shown in Appendix A, demonstrates the visualizations of the Light Recipes described in Table 1. They show the contribution of each wavelength of light (in nanometers, nm) on the *x*-axis to the overall recipe. The *y*-axis shows the Photon Flux Density (PFD; µmol/m^2^/s) per wavelength of light (nm). For the four OSRAM Light Recipes, the *y*-axis was fixed at 7 µmol/m^2^/s for comparison across Light Recipes. The 24VHELED, having around 5.86 times less PFD than the OSRAM lights, was set to its own *y*-axis scale of 0.0 to 0.5 µmol/m^2^/s per nm. For our experiments, we designed our OSRAM Light Recipes to be standard vegetative (High Red, derived from Valoya), PAR-dichromatic (No Green), High Blue and Green, and High Far-Red.

### 2.3. Environmental Conditions

To measure and record relative humidity (%) and air temperature (°C), we used two ‘TinyTag View2 Loggers’ (Gemini Data Loggers, Chichester, UK), one located between layers 1 and 3, and the second located between layers 4 and 6, to monitor changes in climate chamber microclimate. We used a ‘HOBO MX CO_2_ Logger’ (Onset, Bourne, MA, USA) to measure CO_2_ in our climate chamber; the average CO_2_ was 448.84 ppm over the experimental periods and was uncontrolled. Table 2 details the information collected from these devices for the first and second iterations (Period One and Period Two) of our experiments: they show the Low and High Zone Temperature (°C), the Low and High Zone Relative Humidity (%), the Low and High Zone Vapor Pressure Deficit (VPD; kPa), and the Climate Chamber temperature (°C). We set our temperature to be 21 °C and the RH to be 65% in the climate chamber with the aim of meeting VPD targets of around 0.85 kPa, which is advantageous for plants in the phenologically young propagation stage, such as microgreens. Overall, the values in Table 2 show that we achieved stable, targeted environmental conditions.

For Fertilizer, we used Pioner NPK (Mg) Macro Blue 14-3-23 (3) for our macronutrients (Pioner, Herlev, Denmark); for our micronutrient Fertilizer, we used Pioner Micro with Iron (Pioner, Herlev, Denmark). For the concentrated Fertilizer solution, we used 10 kg of Macro per 100 L of water and 1 L of Micro per 100 L of water. For use in our hydroponic system, this was applied at a 15% dilution for a rate of 85 mL of concentrated nutrient solution per 10 L of water. In addition, we used Nitric Acid to control the pH with a Nutrient Solution (NS) pH target of 6.0. We set our pH for the Fertilized treatments due to the necessity of ensuring that the nutrients were available for use by the plants. Because this was not a problem for the Unfertilized treatments, we did not control the pH as rigorously. For Electrical Conductivity (EC, mS/cm), we did not modify it after achieving an initial value with a Fertilizer of around 1.8 mS/cm, according to our Nutrient Solution dilution. For measuring pH and EC, we used a “Mesur Hand Instrument” (Senmatic DGT-Voltmatic, Søndersø, Denmark), which has a sensitivity for pH of 0.1 and a sensitivity of 0.01 for EC values under 2.5 mS/cm. Overall, Table 3 shows that we had consistent conditions across our different experimental periods for both pH and EC.

### 2.4. Measurements and Calculations

For each generation of microgreens, 8 days after sowing, the entirety of the above-substrate plant biomass was harvested, with each cup as a replicate (*n* = 3 per species). Fresh weights (FW) were collected for each complete cup using a gravimetric scale with a sensitivity of d = 0.01/0.001 g (PG503-S DeltaRange, Mettler Toledo, Columbus, OH, USA). After the total FW was taken, the microgreens were placed in a drying oven for four days at 60 °C [59,60]. The dry weight (DW) was then taken for each replicate using the same scale. Dry weight percent (DW%) was then calculated as:(1)$$Dry\;Weight\;Percent=\frac{Dry\;weight}{Fresh\;weight}\times 100$$

Heights were taken using a ruler (±1 mm) at three locations for an average plant canopy height. Leaf area (cm^2^) measurements were taken using an image recognition algorithm that inputted images of harvested leaves on a clean, neutral background and a reference frame object of known surface area to compare the spatial footprint of the leaves to. Ten leaves were randomly selected per replicate. The estimated Leaf Area Index (LAI) was calculated as seen below.
(2)$$LAI=\frac{Average\;leaf\;area\;\left({cm}^{2}\right)\;per\;stem\;\times \;Number\;of\;stems\;per\;cup}{Spatial\;footprint\;of\;cup\;({cm}^{2})}$$

To measure our total Climate Chamber energy use in kilowatt hours (kWh), we used the utility power meter housed in our growing facility. We took measurements every hour on measurement days to obtain baseline values (± 0.1 kWh). We calculated energy use efficiency (EUE; g FW/kWh) at two resolutions: only the light energy use over the four-day experimental period and total energy use (light energy + total system) using the following equations: We also measured Light Use Efficiency (g FW/PFD) to demonstrate differences between lights.
(3)$$Light\;EUE=\frac{grams\;fresh\;weight}{kWh\;of\;light}$$
(4)$$Total\;EUE=\frac{grams\;fresh\;weight}{Total\;kWh}$$
(5)$$LUE=\frac{grams\;fresh\;weight}{PFD}$$

Water use was measured according to flows from our reservoir. We conducted an experiment by taking height measurements for every liter of water added from 0 to 37 L. With these data points, we constructed an exponential equation; inputs (x) were water height in centimeters ± 1 mm, which predicted Liters of H_2_O with an R^2^ of 1.00. Therefore, we were able to predict the amount of total evapotranspiration in our system by measuring heights at 3 separate locations for an average value in cm (±1 mm). From this information, we determined the Water Use Efficiency (WUE) for our different experiments, which, given our measurements, shows total evapotranspiration as our Water Use metric.
(6)$$WUE=\frac{grams\;Fresh\;Weight}{Liter\;H_{2}O}$$

Land Surface Use Efficiency (SUE; grams FW/m^2^/day) was calculated to show the density and speed of production. This was carried out by transforming ‘grams/cup’ to ‘grams/m^2^′ using the dimensions of the growing surface area as a percent of 1 m^2^. Values were then multiplied by this factor and then divided by the growing period (in days), as seen below.
(7)$$SUE=\frac{\frac{grams\;Fresh\;Weight}{m^{2}}}{\#\;of\;Days}$$

We also calculated the cost to contextualize how much, in dollars, it costs to produce a unit of fresh microgreen biomass. We calculated this using operating costs (energy, water, materials such as cups and pads, and Fertilizer); labor costs were not calculated, and we also did not include up-front capital costs of the lights, for example. Each case was calculated for one layer of use (generation) at our experimental density of 18 cups. We incorporated the previously calculated energy and water use values from previous measurements and multiplied them by the cost per unit, in US dollars, of the Danish market at the time of analysis in the fall of 2023.
(8)$$CUE=\frac{grams\;Fresh\;Weight}{Dollar\;Operating\;Costs}$$

### 2.5. Statistical Analysis

Data were analyzed with R [61] and R Studio (R Studio, Boston, MA, USA). We fitted a linear model and used estimated marginal means [62] and 95% confidence intervals to determine significant differences between Light Recipes at every treatment combination using Tukey’s Honestly Significant Difference (HSD); results were significantly different at *p* ≤ 0.05. For the linear model, we used Box-Cox analysis to identify the response transformation from 0 (logarithm) to 2 (square power) in steps of 0.5. We then fitted our initial model and did a backward model selection based on Akaike Information Criterion (AIC) scores. After making our model reduction hypothesis (Type III test) for the model, we extracted studentized residuals and fitted values from initial models and normal quantile plots to test for normality of the data, which were verified. Separate models were generated for each species (kohlrabi, mustard, and radish); they had the same fixed effects of Light Recipe, Seeding Density, and Fertilizer, with all higher-order interactions tested. These models were used for our primary analyses: FW and DW (kg/m^2^) in Figure 2, and for the FW and DW linear model tables of outputs (Appendix A) and the Resource Use Efficiency figures in Appendix A. Other response variables were analyzed separately in R, such as a Multi-way ANOVA (Table 4) with the additional fixed effect of Species; other regressions were made in Excel. We also used R to compose Pearson’s Correlation [63] tables for each species (seen in Appendix A in Appendix A) and conduct Principal Component Analyses (PCA).

## 3. Results

### 3.1. Microgreen Biomass Production and Morphology

Table 4 shows the multiway Type I ANOVA results for the complete interspecies dataset. All of the main effects were significant for FW and DW (*p* < 0.001), with one second-order interaction for FW Fertilizer by SD (*p* = 0.02). We used eta-squared (η^2^), which describes the proportion of each model effect’s sum squares to the total sum squares, to demonstrate the effect size of the source of variation’s explanation of variance within the ANOVA. The results in Table 4 demonstrate that the most variation was explained by the ‘Species’ effect (65.28%), with ‘Fertilizer’ having the second highest for FW (15.65%), and ‘Species’ and ‘SD’ having the highest and second highest for DW (77.24% and 10.28%, respectively). This shows that there was a large amount of variation explained by Species, which is expected, followed by Fertilizer, SD, and Light Recipe. Furthermore, to evaluate the potential influence of other parameters beyond our experimental treatment factors, we ran a series of linear models for each species (kohlrabi, mustard, and radish) at each Fertilizer level with their SD averaged for both FW (kg/m^2^) and DW (kg/m^2^). Our independent variables were pH, CO_2_, Vapor Pressure Deficit, Temperature, and Relative Humidity; our dependent variables were FW and DW. We found no significant effect on either FW or DW via environmental parameterization at any treatment level combination, which was likely due to the low degree of variability in our experimental conditions. For DW, the average *p*-value of our effects was 0.61. For FW, the average *p*-value was 0.65.

In order to visualize the distinct qualities of our microgreen species of interest (kohlrabi, mustard, and radish), we conducted a PCA (Figure 1) with our primary dependent variables: FW (kg/m^2^), DW (kg/m^2^), DW % (pct), and resource use efficiencies: Energy Use Efficiency (EUE), Light Use Efficiency (LUE), Water Use Efficiency (WUE), land Surface Use Efficiency (SUE), and Cost Use Efficiency (CUE). The left panel shows the Unfertilized PCA, with PC1 explaining 77.91% and PC2 explaining 14.44%, for a total of 92.35% variance. The right panel shows the Fertilized PCA, with PC1 explaining 75.94% and PC2 explaining 14.21%, for a total of 90.15% of variance. From these PCA plots, we can see that there was a very consistent response across fertilization levels. Furthermore, the PCA plots show the distinct grouping of our species in response to our experimental conditions. We can also see that the same species orientation is generally seen across fertilization treatments, with similar directionality and length of the loadings of the different dependent variable responses.

Figure 2 below shows the mean DW (kg/m^2^) and FW (kg/m^2^) ± 95% confidence intervals for kohlrabi (top), mustard (middle), and radish (bottom) microgreens. The letters above each Light Recipe show the compact letter display (CLD) from Tukey’s Honestly Significant Difference (HSD); they compare significant differences between Light Recipes within each treatment. For kohlrabi FW, we can see that the HFR recipe produced the most biomass; it was significantly greater than both the 24VHELED and the NG Light Recipes. The other recipes were generally not significantly different from each other. For DW, the results were generally the same as for FW, with the HFR being significantly greater than all other Light Recipes except for the HR. Furthermore, the 24VHELED for DW was significantly less than all other Light Recipes at every treatment combination. Our linear model output showed that the kohlrabi FW had an adjusted R^2^ = 0.90 and the DW had an adjusted R^2^ = 0.87. For mustard FW, we can see that the HFR recipe also produced the most biomass; it was significantly greater than every other Light Recipe at every combination of SD and Fertilizer level. The other Light Recipes were not significantly different from one another. For DW, the results were the same as for FW. Additionally, mustard FW had an adjusted R^2^ = 0.85, and the DW had an adjusted R^2^ = 0.84. For radish FW, the HFR Light Recipe produced the most biomass, and it was significantly greater than the 24VHELED, HBG, and the NG recipes across all combinations of Fertilizer and SD for FW. The 24VHELED was once again significantly lower than all other Light Recipes across all experiments for FW. For DW, radish accumulation was more dynamic than kohlrabi and mustard, with the HR recipe having the significantly greatest amount of DW without Fertilizer. However, with Fertilizer, only the 24VHELED was significantly less than the other Light Recipes. Our linear model output showed that the radish FW had an adjusted R^2^ = 0.86 and the DW had an adjusted R^2^ = 0.92. For FW in general, as the ANOVA table shows above, we can see a very large effect of Fertilizer on biomass accumulation (across panels), while for DW, we can see that SD contributed more to variation than Fertilizer, although there were still improvements due to this. Averaging the species together, we saw that Fertilizer improved biomass by a factor of 1.63 (24VHELED) vs. 1.56 (OSRAM). On average, across species, the higher intensity OSRAM lights produced 1.10 times the FW biomass compared to the 24VHELED and 1.15 times the DW biomass averaged across species.

Figure 3 shows the association between different Light Recipe components (µmols/m^2^/s) and FW biomass accumulation (g/cup) for only the four OSRAM Light Recipes that had the same PFD (270 µmols/m^2^/s). These values are shown averaged across Fertilizer for the sake of clarity, as there was no interaction between Fertilizer and Light Recipe (Table 4). Within each panel, all upper values, a shade lighter than their corresponding symbols, indicate the High SD, while the lower, darker shapes indicate the Standard SD. This figure shows that, regardless of other factors, adding more Far-Red light improved biomass even though the HFR recipe had 99% of the PFD as the lower Far-Red Light (LFR) Recipes. For instance, kohlrabi HFR had 121% of the FW biomass of the LFR recipes, mustard had 127% of the FW biomass of the LFR recipes, and radish had 110% of the FW biomass of the LFR recipes. The R^2^ values ranged from 0.62 at the lowest up to 0.99 at the highest, with an average R^2^ of 0.76. Radish had the best fit, with kohlrabi having the worst. The next row down shows the association between Red light and FW biomass accumulation, where there was a highly negative correlation, with the lowest R^2^ being 0.47, and the highest being 0.97, with an average of 0.83. The third row down shows Green light from FW biomass. Interestingly, the No Green (NG) recipe had 1.02 times more PFD but was generally less productive than the higher Green recipes for FW. For instance, for kohlrabi, the NG only had 90% of the FW biomass compared to the Green-containing recipes, while for mustard, the NG had 93% of the FW biomass of the Green-containing recipes, and for radish, the NG had 99% of the FW biomass. This linear association was the weakest of our Light components, but there was still a positive association where, on average, the Light Recipes that received Green light were greater than the Light Recipe without Green Light. The last row shows Blue light by FW, where we found a consistent positive association with biomass up to around 55 µmols/m^2^/s, after which it decreased. This is shown by our quadratic regression fitting, with R^2^ ranging from a low of 0.33 up to a high of 0.92, with an average of 0.75.

Furthermore, Figure 3 shows how the response differed between SDs; that is, we hypothesized that a higher SD (with higher corresponding LAI) would respond more positively to Green and Far-Red light with its lower absorption and more negatively to Red and Blue light. In every case, except for the radish Green light by FW, this was consistently shown, with the Far-Red and Green High SD having a greater slope and the Red having a more negative slope; the Blue was quadratic, but we can see that the exponent in the quadratic equation was more negative for kohlrabi and mustard High SD, but not so for radish. This only shows correlation, but it is especially relevant for microgreens, compared to other plants with lower LAI values, to see that both Far-Red and Green Light were positively associated with biomass, while Red was negatively associated with biomass, and Blue showed a distinct quadratic shape. This could be an important justification for using Green or Far-Red light for microgreen production environments where light is highly in demand because of competition from high LAI and leaf absorption due to the competitively high seed sowing densities.

As shown in Figure 3, Far-Red light was associated with improvements in FW. Some reasons for this can be contextualized in Figure 4, which shows the association between the amount of Far-Red light (µmols/m^2^/s) and height (cm) improvements for all three microgreen species. The panel on the left shows the Unfertilized treatments, and the panel on the right shows the Fertilized treatments. Within each figure, red circles represent kohlrabi, blue triangles represent mustard, and black squares represent radish. Furthermore, linear models were conducted for each species at their respective treatment levels, and their significance is labeled per species within each figure. Overall, Figure 4 shows that more Far-Red light improved height for all species across Fertilizer treatments: for radish without Fertilizer, the R^2^ was 0.98, and with Fertilizer the R^2^ was 1.00. For mustard without Fertilizer, the R^2^ was 0.96, and with Fertilizer, it was 0.60. For kohlrabi without Fertilizer, the R^2^ was 0.98, while with Fertilizer, the R^2^ was 0.68. Interestingly, the mustard responded the most strongly to additional Far-Red light, especially without Fertilizer, as its height improved greatly, being even greater than the radish’s. These figures show that, for kohlrabi, Far-Red light was associated with height improvements (lowest Far-Red height divided by highest Far-Red height) by about 1.41 times without Fertilizer, and 1.29 times with Fertilizer; for mustard, about 1.47 times without Fertilizer, and 1.24 times with Fertilizer; finally, for radish, around 1.40 times without Fertilizer, and 1.25 times with Fertilizer. Furthermore, for kohlrabi, these figures show that Fertilizer overall improved height by around 1.24 times compared to Unfertilized treatments; for mustard, 1.23 times; and for radish, 1.26 times. Therefore, we can see that, for all three species, height was overall improved via the application of Fertilizer by a factor of around 1.24. Furthermore, within Fertilization levels, we can see that there were greater height differences between the lowest and highest Far-Red rates for the Unfertilized microgreens compared to Fertilized treatments. Interestingly, the microgreen slopes were nearly parallel, indicating a stable response of height to changes in increasing Far-Red light across species.

Figure 4 also shows the association between the amount of Far-Red light (µmols/m^2^/s) and leaf weight (g leaf FW/plant), leaf area (cm^2^), and leaf area index (cm^2^/cm^2^). For all three response variables, we saw a clear positive linear relationship, except for Fertilized mustard. This was largely due to the high values of the lowest Far-Red light amounts, which responded strongly to Fertilizer application, and therefore we did not see a positive or negative response. For leaf fresh weight, these figures show that Far-Red was generally associated with higher leaf biomass for both Fertilized and Unfertilized microgreens by a factor of 1.91 without Fertilizer, and 1.29 with Fertilizer across species. In general, Fertilizer improved leaf weight by a factor of 1.77. For leaf area, these figures also showed improvements associated with Far-Red light for both Fertilized and Unfertilized microgreens by a factor of 1.73 without Fertilizer, and 1.23 with Fertilizer. Furthermore, Fertilizer improved leaf area by a factor of 1.74. For estimated LAI, the same dynamic was seen for both Fertilized and Unfertilized microgreens by a factor of 1.74 without Fertilizer, and 1.25 with Fertilizer. In general, Fertilizer improved LAI by a factor of 1.74.

Table 5 shows that although there were clear positive associations between Far-Red light for height, leaf area, LAI, and leaf FW, we saw a more dynamic response of Far-Red light with SLA, which is not surprising. We saw a clear effect of overall improvements comparing the 24VHELEDs to the OSRAM, but within the OSRAM, we saw a consistent pattern where the amount of leaf area improved more than the amount of leaf DW at higher levels of Far-Red light.

Figure 5 below shows regression plots for height by leaf FW, leaf DW, leaf area, and FW. The above figures showed that Far-Red light was positively associated with many improvements in microgreen morphology. These changes led to improvements in height and leaf area, for instance, which likely improved light interception and incident light. Therefore, adding Far-Red light was associated with morphological improvements, which in turn were correlated with increased biomass. This can be seen in the bottom-right figure, which shows that changes in height were associated with improvements in FW biomass accumulation for all three species. In general, kohlrabi (red circles) had the greatest fit, with the highest R^2^, while mustard (blue triangles) generally had the worst fit.

Figure 6 shows the effect of increasing light integral on microgreen biomass in units of Cumulative Light Integral (CLI; mol/m^2^), which is the PAR Daily Light Integral (DLI; mol/m^2^/day) times the number of days exposed to light. The leftmost points are the low-intensity 24VHELED; the middle four are our regular OSRAM recipes; and the two rightmost points are higher CLI High Red OSRAM recipes derived from early parameterization experiments. Figure 6 shows that there are similar effects across species for increasing CLI and its effect on FW and DW; that is, increasing light integral had a much greater negative impact on FW than on DW, the latter of which was relatively stable. Our quadratic curves show that there is a middle point around 45 mol/m^2^, past which gains decrease and even become negative. However, this is modified by the Light Recipe. This ‘optimal’ middle point was more pronounced for FW than it was for DW. In general, we can see that the smaller the microgreen, the greater its sensitivity to increasing light integral for FW. Furthermore, we can see that radish had a much flatter DW curve with a higher R^2^ compared to the other species, indicating a smaller response to Light Recipe and a closer fit with increasing light integral. Kohlrabi and mustard had a greater deviation (lower R^2^), with some Light Recipes performing better than others. These figures also demonstrate the benefit of Far-Red light, as it was the point generally high above the curves.

Table 6 below shows the average dry weight percent (DW%) for each species at the level of Light Recipe and fertilization, with results being averaged across Standard and High SD. For kohlrabi without fertilization, the HR recipe had the highest DW%; with Fertilizer, the highest was the NG Light Recipe. Overall, the 24VHELED had the lowest DW% for kohlrabi, which is not surprising considering it had around 5.86 times less light (µmols/m^2^/s) than the OSRAM Light Recipes. For mustard without fertilization, the NG recipe had the highest DW%, and the lowest was the 24VHELED. With Fertilizer, the highest was HR, and the lowest was HFR, likely due to the large accumulation of FW in the HFR recipe. For radish without fertilization, the HR recipe had the highest DW%, and the lowest was the 24VHELED. With Fertilizer, the highest DW% was NG, and the lowest was HBG. From this table, for all species, we can clearly see that Fertilizer primarily impacts FW accumulation and does not influence DW accumulation to the same degree. Overall, mustard had the highest average DW% (6.36%), with radish having the next highest (6.18%), and kohlrabi having the lowest DW% (5.92%).

Furthermore, it is important to consider how the inherent life history traits impact the biomass results. For instance, kohlrabi had the lightest seed weight (approximately 0.16 g/100 seeds), mustard had the middle seed weight (approximately 0.72 g/100 seeds), and radish had the heaviest seed weight (approximately 1.32 g/100 seeds). Figure 7 shows the FW (left panel) and DW (right panel) averaged across Light Recipe and Seeding Density, with categories for Fertilized (black) and Unfertilized (red). Therefore, kohlrabi shows the leftmost, mustard shows the middle, and radish shows the rightmost points within each panel, respectively. The figures show that the production of the microgreens fits very closely with the starting seed amount. For FW with Fertilizer, there was an R^2^ = 0.99, and without Fertilizer, the R^2^ = 0.98. However, the slopes show that Fertilizer greatly improved the amount of biomass gained per unit of starting seed, with a slope 1.54 times greater than Unfertilized treatments. For DW, this relationship was less pronounced, with nearly parallel slopes (Fertilizer having only around 1.06 times the slope of Unfertilized treatments). With Fertilizer the DW R^2^ = 1.00, and without Fertilizer the R^2^ = 0.98. Figure 7 also shows that mustard was on average below the regression line. Overall, Figure 7 shows that radish produced around 3.80 times the FW biomass compared to kohlrabi and around 3.86 times the DW of kohlrabi. Mustard produced around 2.47 times the FW compared to kohlrabi and around 4.48 times the DW of kohlrabi. Finally, the radish produced around 1.53 times the FW biomass compared to the mustard and around 1.49 times the DW biomass. These values are generally the same, proportionally, across the Light Recipes and Fertilization levels.

Table 7 shows that averaging across all three species, Fertilizer improved FW by a factor of 1.60. Fertilized kohlrabi was on average 1.58 times greater, mustard was 1.66 times greater, and radish was 1.56 times greater for FW than Unfertilized microgreens. Interestingly, for DW, there was a pattern whereby there was a falling off of Fertilizer’s effect, diminishing with the larger microgreens. For instance, kohlrabi Fertilization improved DW biomass by 1.36, mustard by 1.25, and radish by 1.11 compared to their Unfertilized microgreen counterparts. Overall, Fertilizer improved DW by a factor of 1.25. This is a common pattern whereby we found that fertilization generally increased FW biomass production more than DW; this difference was significant for kohlrabi (t-test; *p* < 0.01), mustard (*p* < 0.001), and radish (*p* < 0.001) comparing FW to DW.

We also investigated how the degree of fertilization could help define production improvements. Figure 8 below shows the influence of fertilization at different dosages (0 days = 0; 4 days = 0.5; and 8 days = 1) for the HR and 24VHELED Light Recipes (red and black, respectively). Although we saw some variation in FW accumulation (R^2^ = 0.84 to R^2^ = 0.99), there was a remarkably consistent linear accumulation of DW for both Light Recipes (R^2^ = 0.95 to R^2^ = 1.00). Interestingly, as is consistent with our findings, Figure 8 shows that although the FWs were close for the different Light Recipes of very different light intensity (46 μmol/m^2^/s vs. 270 μmol/m^2^/s), the DW accumulation was relatively greater for the high-intensity lights compared to the low-intensity ones. Overall, this figure demonstrates that we did not saturate the microgreens with Fertilizer, as gains were linear and did not drop off, and that there were biomass improvements for both the initial four days of germination and the four days under lights.

Table 8 shows the improvements to biomass production across Light Recipes due to increasing SD. This was carried out by dividing the High SD biomass values by the Standard SD (grams/cup). Table 8 shows that the High SD never reached the 1.50 benchmark, as they are sown with 50% more seeds, indicating some loss of per-plant production. For kohlrabi, adding Fertilizer improved this ratio for FW; however, using a t-test, we found no significant differences between Fertilizer levels. For mustard, in opposition to the kohlrabi, adding Fertilizer did not improve this ratio. This is because the Standard SD with Fertilizer had a relatively higher amount of production compared to the High SD with Fertilizer. Without Fertilization, for mustard FW, this ratio was almost 1.50 (1.45, Table 8). Using t-tests, we found significant differences in FW between Fertilization treatments (*p* = 0.02). Radish, also in opposition to kohlrabi, showed that adding Fertilizer did not improve the SD biomass ratio for FW. We found no significant differences between fertilization levels. In general, we found that kohlrabi was 91% efficient, mustard was 92% efficient, and radish was 91% efficient at High SD compared to Standard SD. Inversely, we can also calculate that, on average across species, FW production was 37% more spatially efficient at High SD. Overall, Table 8 shows that High SD saw some loss in relative production. Therefore, there is a tradeoff between biomass produced per plant and spatial use efficiency (per m^2^).

### 3.2. Resource Use Efficiencies

Table 9 shows the mean RUE ± SE averaged across species for each Light Recipe with and without Fertilizer; lowercase letters show significant differences between Light Recipes within each Fertilizer level. This was carried out to show the general RUE of the system as a whole, growing kohlrabi, mustard, and radish microgreens. Species-specific results can be seen in Appendix A. For LUE (g FW/PFD), the 24VHELEDs were always the most efficient, while the HFR was the second highest with and without Fertilizer, which was significantly greater than the other 3 OSRAM Light Recipes without Fertilizer. Overall, Fertilizer improved LUE by a factor of 1.26. In general, across Fertilizer levels, the 24VHELEDs had a greater LUE by a factor of 4.86. For Light EUE (g FW/kWh of light), the NG was the most efficient given its much lower power use compared to the other OSRAM Light Recipes, likely due to the highly efficient photon conversion efficiency of the Blue and Red band LEDs. Even though the NG was a lower-performing recipe for biomass, it had a significantly higher LUE than other OSRAM Light Recipes. In general, the HBG recipe was a good producer of biomass but was energetically expensive, and it was significantly lower than the HFR and the NG recipes across treatments for Light EUE. Overall, Fertilizer improved Light EUE by a factor of 1.66. In general, across Fertilizer levels, the 24VHELEDs had a greater Light EUE by a factor of 5.11. However, light was not our only energy consumer; our climate chamber also consumed a very large proportion of the total energy (around 99%). This was likely due to the age of the climate chamber itself and its method of precisely controlling the atmospheric conditions. We therefore separated this EUE (total) from the aforementioned EUE (lights), as it is not necessarily representative of producers or even other researchers. The total EUE showed that although there were differences between the light use efficiencies, the overarching pattern of biomass production seen in the above section is reproduced here. As such, the HFR recipe performed best, and the 24VHELED performed slightly better due to its EUE advantages. For Total EUE (g FW/total kWh), the production advantages of the HFR showed through, as this recipe was significantly greater than the other 3 OSRAM Light Recipes at both Fertilizer levels. Overall, Fertilizer improved Total EUE by a factor of 1.67.

For WUE (g FW/L H_2_O), a similar pattern was seen where the HFR and the 24VHELEDs were the most efficient recipes and were significantly greater than the HR and NG recipes, while only the HFR was greater than the HBG. In general, for WUE across Fertilizer levels, the 24VHELEDs were greater by a factor of 1.04. Overall, Fertilizer improved WUE by a factor of 1.67. Looking at SUE (g FW/m^2^), the HFR was the significantly most efficient recipe, while the 24VHELED was the significantly least efficient, followed closely by the NG recipe. Overall, Fertilizer improved SUE by a factor of 1.67. In general, across Fertilizer levels, the 24VHELEDs were only 88% as efficient as the OSRAM LEDs for SUE. However, this ignores the cost of the lights to get there. Therefore, it is also important to consider CUE (g FW/Dollar), where the 24VHELEDs were the significantly most efficient producers, while the HFR was the only recipe significantly greater than the other OSRAM recipes. Overall, Fertilizer improved CUE by a factor of 1.65. In general, across Fertilizer levels, the 24VHELEDs were greater by a factor of 1.22 for CUE. Considering the cost, with the OSRAM lights producing on average 1.07 times the FW without Fertilizer, and 1.10 times the biomass without Fertilizer as the 24VHELEDs, it would take 427 generations and 299 generations, respectively, to make up for this cost deficit, assuming each cup of microgreens sold for 3 dollars and there were 80 cups (full capacity) per layer. For DW, this is even more pronounced. These RUE values demonstrate the tradeoffs inherent in microgreen production. Although the higher-intensity, programmable OSRAM LEDs performed the best for biomass, there are other dynamics to consider when selecting production methods. Perhaps once the efficiency of Green and Far-Red LED bands reaches parity with Red and Blue, and the costs of high-quality programmable LEDs are lowered, the RUE values will change. For now, as is common practice for growers, cheap and easily accessible LEDs produce lower, but competitive, amounts of microgreens with lower energy and cost inputs per unit output.

## 4. Discussion

### 4.1. Effects of the Light Spectrum on Microgreens

In this experiment, we showed that microgreens grown for four days in the dark and for four days in the light had significant responses to different fertilization regimes, seeding densities, and types and intensities of light. Our experimental results showed that the best Light Recipe was High Far-Red for FW and DW biomass production, which was also strongly associated with morphological improvements in height, leaf area, LAI, SLA, and leaf weight. Overall, our results showed the advantages of engineering a shade-avoidance response using a low R:FR ratio (approximately 1.51) paired with high-intensity lighting (measured at 270 µmols/m^2^/s) that includes Green light, as well as applying Fertilizer at High SD. The advantages of Far-Red light shown in this study are not surprising considering recent research on the effects of Far-Red light on plant growth and its positive impacts on plant physiology and morphology [15,31,37,48,49,50,64,65,66,67]. For instance, low Red:Far-Red ratios have been shown to induce shade avoidance responses, which can manifest in changes in leaf expansion and height [64,68]. This occurs via the Red:Far-Red ratio, which converts the biologically inactive red absorbing form (Pr) of phytochrome to the biologically active far-red absorbing form (Pfr), which then can cause downstream shifts in gene expression for specific responses to the environment, such as a shade-avoidance [69]. This has also specifically been shown for microgreens; for instance, studies have shown that adding Far-Red light directly led to improvements in height, leaf area, FW, and DW [14,70,71]. Furthermore, a study on microgreens found that adding small amounts of Far-Red to a pure-Blue recipe improved height and stem extension rate (SER) compared to UV + Blue, pure Red, and pure Blue Light Recipes [72]. Another study found that adding Far-Red light, thereby decreasing the PPE to around 0.60, which optimized both leaf area and hypocotyl length, significantly increased the SER of arugula, cabbage, kale, and mustard microgreens [73]. Our HFR recipe had a PPE of 0.75, which was the lowest of all the recipes tested in this experiment. Therefore, because the responses between Far-Red light incidence and height, leaf area, leaf weight, and biomass were linear, we likely could have included more Far-Red light; this is a place for further research to identify at which point improvements associated with Far-Red taper off.

Beyond morphological changes, there are also improvements to photosynthesis offered by Far-Red light; this has been shown in previous research in greater detail [48,49,64,68,74,75,76]. Although we did not take direct measures of photosynthesis, such as via gas exchange, we did measure dry weight, which is the result of anabolic processes downstream from photosynthesis. It has been suggested that it is useful to incorporate Far-Red light in a similar proportion to that seen in natural full sunlight, under which conditions plants evolved: around 1 µmols/m^2^/s of Far-Red light to every 5 µmols/m^2^/s of PPFD, at a ratio of 0.19 [48]. In our study, we had around half this ratio at 0.09 for three of our OSRAM Light Recipes, and around 0.30 for our best-performing HFR recipe. This ratio of 0.30 likely indicates that we engineered a shade-avoidance response with our Light Recipe, which could explain our positive results for morphology and biomass with Far-Red light. Importantly, Far-Red light has been shown to offer improvements to overall canopy photosynthesis up to 40% Far-Red in a Light Recipe [53]; our HFR recipe had a Far-Red component of around 23%. Therefore, further research could test whether gains increase beyond 23% up to 40% for microgreens as well. Previous research on microgreens and other plants supports this. For instance, a Light Recipe with only 7% Far-Red light was shown to increase height and FW in kohlrabi and mizuna microgreens compared to a dichromatic Blue-Red recipe [77]. In another experiment, it was shown that mustard microgreens exposed to Far-Red light also increased yield, but the response shown in their study was not as strong as that seen with ours [71]. This could be because of our more dynamic Light Recipe (e.g., more Far-Red and Green) compared to theirs. Although we did not specifically measure photosynthesis, we saw the highest FW and DW biomass values with the HFR Light Recipe, the latter of which indicates downstream improvements. These improvements could be due to improved photosynthesis, but they could also be the result of morphological changes that improved light interception, which offered more opportunity for photosynthesis. The SLA results indicate that the leaf DW did not increase as much as the leaf area did, indicating that it is likely a combination of the two. It is also interesting to point out that Far-Red light did not shift allocation patterns in a zero-sum manner but instead improved photosynthetic processes, resulting in more overall biomass. For instance, one study showed there was no loss in the production of root biomass when total above-ground FW and DW biomass were improved [78]. We also conducted some exploratory work on radish microgreens and found no loss in root biomass with the HFR recipe compared to the other OSRAM recipes. This avenue of investigation is important, as recent research has noted that improvements in plant production must likely come from changes in photosynthetic efficiency, as other avenues, such as harvest indices, have been pushed close to their theoretical maximum [79,80].

Although this experiment and past research have shown the benefits of using Far-Red light, it is important to note that these improvements are dosage, wavelength, and synergistically dependent. As previously mentioned, the dosage of Far-Red was found to be effective up to 40% of a Light Recipe [53]. Additionally, Zhen et al. demonstrated that photons with a wavelength > 752 nm are not as effective as those between 700 and 751 nm, and secondly, that the overall effect is dose-dependent: past a certain point, gains are not made as photosystems can become imbalanced [48]. The synergy requirements of Far-Red have been described through the lens of the Emerson Effect hypothesis [38]. The Emerson Effect describes how plants exposed to light both at 680 nm and light past 680 nm (Far-Red) had improved rates of photosynthesis compared to those with no long-wave photons. Although we found positive associations with Far-Red light at several doses, we did not test Light Recipes with pure Far-Red or Far-Red knockout. However, previous work on Brassica microgreens that applied pure Far-Red and Blue light at night in addition to day-time ambient Greenhouse sunlight showed that the Blue light and Dark (control) nighttime treatments resulted in higher-quality microgreens compared to the pure Far-Red treatments [81]. Furthermore, beyond these dynamics, another study showed that the benefits and improvements offered to biomass and morphology were accumulative in nature; they found that a late harvest of 25 days compared to 16 days was significantly advantageous for lettuce [50]. Therefore, the growing period is also a fourth component to consider in Far-Red light research, as the benefits of the HFR Light Recipe shown in this study could likely be even more pronounced given a longer experimental growing period. This is an area of research that demands more attention, as it has been noted that limited studies have been conducted on the influence of Far-Red light on the morphological and photosynthetic parameters of plants [66].

Much previous research has focused primarily on Blue-Red dichromatic recipes, or even monochromatic Light Recipes, and their differing proportions [82,83,84,85,86,87,88,89,90]. However, it is increasingly becoming clear that it is important to view the activity of light through the lens of the Emerson ‘Enhancement’ effect, which shows the synergistic effects of light on plants and necessitates broader-range Light Recipe inclusions [49,74,91,92]. In line with this, we showed that the NG recipes were negatively associated with microgreen production in our experiment compared with the other more dynamics Light Recipes. This is not surprising considering that Green light has been associated with cueing plants’ circadian rhythms, which is essential for appropriate physiological activities [93], improving carbon assimilation [94], and impacting Nitrate assimliation genes (NR and NiR), which can contribute to improved protein or chlorophyll synthesis [95]. Furthermore, Green light is useful in tandem with Blue light for regulating height, as the Blue:Green ratio is important for Cryptochrome-mediated changes in height [96]. Our results are echoed by experiments that have shown that a Light Recipe with 18% Green vs. no Green (pure dichromatic Blue and Red) for mustard microgreens resulted in an increase in hypocotyl length and shoot FW [77]. In a similar vein, another study found that adding only 6% Green light improved stem extension rate and height compared to pure control, Red, and pure Blue recipes for cabbage, kale, and mustard microgreens [97]. Another study showed that higher levels of Green light increased root growth, overall biomass accumulation, and photosynthetic rate in lettuce at higher PFDs [98]. Furthermore, another study on 20 different varieties of Brassica microgreens found that Green light-containing recipes offered significant improvements to both vegetative growth and morphology [45]. Beyond the positive associations with Green light in a general sense, the conditions of producing microgreens might not favor the use of too much Red and Blue light, which we showed were both negatively associated with biomass. This could be because, although plants have been shown to absorb around 90% of Red light (600–700 nm) and Blue light (400–500 nm), their translation into photosynthesis is not necessarily as high, with much of the light being potentially lost as nonphotochemical quenching in the form of heat [42]. Therefore, Green light and Far-Red light could be beneficial for competitive environments where these spectral bands, with their lower absorption, can pass through canopy leaves and still be utilized by sub-canopy leaves, as well as cueing photomorphogenesis in the same manner. We found this to be consistent with our results, as the PAR-dichromatic (NG) Light Recipe, which had a very similar B:R ratio (0.61) compared to both the HBG (0.66) and the HFR (0.57), was outperformed by a factor of 1.08 for FW and 1.03 for DW. It is also worth noting that the 24VHELEDs performed very well, even though they had around 46% of their Light Recipe composed of Green light.

### 4.2. Microgreen Fertilizer and Seeding Density

Regarding the effect of fertilization on Brassica microgreens in our study, we found that there were significant improvements for both FW and DW biomass production. This was much more pronounced overall for FW than for DW. We have shown that there was around a 1.4 to 2.0 times improvement in FW and around a 1.1 to 1.5 times improvement in DW accumulation as a result of fertilization, depending on the Light Recipe and the SD. Our results are consistent with other experiments that conducted similar experiments; for instance, one study found that Rocket microgreen FW yield was increased by 47% due to Fertilizer, which is close to our value of 66% [99]. Beyond increases in FW and DW, we also saw an increase in height of around 24% on average for our microgreens due to fertilization. In line with this, a previous study on basil showed that adding Fertilizer increased height; they also showed that Fertilizer application increased the number of leaves, leaf area, FW, and DW, which we also found [100]. Another study found a similar result regarding height, which positively responded to Fertilizer application by a factor of 1.70 from 0% to 100% [101]. And in keeping with our results, they also found a greater increase in FW of a factor of 2.10 from 0% to 100% Fertilizer dosage, while for DW, the increase was only 1.39 from 0% to 100% nutrient solution strength.

Experimenting with Seeding Density is a less common practice for microgreen research; there are many different suggested densities, and they all depend on the species or variety of plants used. In this study, we used the recommended Standard SD supplied by a professional microgreen producer and increased it by 50% for our High SD to observe the productivity results. Overall, we did find a significant influence of SD on FW and DW biomass (Table 4); however, the 50% increase in SD did not correspond to a 50% increase in biomass. On average, we found an increase of around 37% for FW, indicating a drop in production. This nonlinear relationship has been shown in similar research involving young cannabis baby greens, where a 4.4-fold increase in SD resulted in only a 1.85-fold increase in FW biomass, with a 2.29-fold increase in DW [102]. We found a similar dynamic in our results, where there was a greater influence of SD on DW than FW production. Beyond productivity, SD is also important to consider for health reasons, as dense plant conditions paired with high Relative Humidity can result in pathogenic microorganism activity [103,104]. For instance, one study detected harmful *E. coli* and other bacteria in some cases of high-density microgreen production [105]. Although we did not specifically test for it, we found little incidence of visible pathogenic presence even at High SD, indicating stable growing conditions.

### 4.3. Microgreen Resource Use Efficiency

In this experiment, we measured the Water Use Efficiency (WUE), Light Use Efficency (LUE), Energy Use Efficiency (EUE), land Surface Use Efficiency (SUE), and Cost Use Efficiency (CUE) of microgreen production. Overall, we showed that the conditions that improved biomass accumulation came with some higher resource inputs. For instance, we found that the cheaper 24VHELEDs had a higher EUE of lights, LUE, and CUE. On the other hand, they were less productive, therefore they had a lower SUE, Total EUE, and WUE; interestingly, although they used less water overall than the other higher-intensity Light Recipes, their lower production only marginally improved their WUE. The HFR recipe was either second most efficient or most efficient in all cases. This was due to the significant advantages to production that Far-Red light offered. There were some significant differences between Light Recipes, but these were mostly the result of underlying productivity dynamics; the larger difference was seen between light types (24VHELED vs. OSRAM). Previous research has found varying results on the impact of Light Recipe on RUE. For instance, Pennisi found that increasing the ratio of Red:Blue light, i.e., using more Red than Blue, increased the EUE of the vegetables consistently [106]. However, the nutrient solution uptake varied inversely. They found that, in general, the environmental impact of the different Light Recipes favored those that had high amounts of Red light and lower amounts of Blue, although both effects were slightly non-linear, with an optimal ratio being seen around Red:Blue of 3, depending on the species. Our results had Red:Blue ratios of 1.51 up to 5.88, with the optimal for us being 1.75 for the HFR recipe.

Beyond light and energy, we also looked at the SUE, CUE, and WUE of microgreen production. Although SUE is primarily a 2-dimensional measure, we can also discuss it in terms of 3-dimensions in our experimental context. That is, we had SUE of around 250 g/m^2^/day for kohlrabi, 600 g/m^2^/day for mustard, and 900 g/m^2^/day for radish (with Fertilizer); our current system has three layers of lights, with three layers for dark germination; therefore, with a height of the three usable layers at around 105 cm, the system can produce around 714 g/m^3^/day, 1714 g/m^3^/day, and 2571 g/m^3^/day for kohlrabi, mustard, and radish, respectively, if all the layers are running at the same time. Furthermore, given the large increase in material inputs needed for a growing system, using a High SD can be considered ideal, improving SUE as well as other metrics. Concerning CUE, we showed that although the OSRAM lights produced 1.16 times the FW and around 1.22 times the DW, compared to the 24VHELEDS, they initially cost over 190 times more to produce equivalent biomass. Over time, this dynamic would change, as we showed that it would pay off in about 400 generations for FW. Finally, the 24VHELEDs require very little upkeep or labor for installation and use, unlike the OSRAM lights, which are heavy and require careful programming. Overall, we can make a strong case that the 24VHELEDs, which are readily available, require no programming, and are relatively cheap, are an efficient choice for growers. For microgreens that can be grown in the form of a vertical farm, this could be even more efficient, as the 24VHELEDs produce little heat and weigh almost nothing, at around 350 g per layer, compared to the OSRAM lights, which are around 9 kg for each panel (27 kg per layer), which requires special attachments and security for successful use. Another important component of microgreen production is WUE. We showed that the low-cost, low PFD 24VHELEDs had the second highest WUE values, while the overall production benefits of the HFR made it the most WUE recipe. Previous work found that similar to our results, the addition of Far-Red light improved Basil biomass over a dichromatic recipe while also improving WUE and EUE [107]. Furthermore, another study on black kale microgreens showed tradeoffs between production and water availability in an attempt to maximize WUE; they found an optimal value of WUE of 88 g/L, while we found equivalent values ranging from a low of 46 g/L up to a high of 94 g/L [108]. Previous research has expanded on this, showing that for broccoli microgreens, around 158–236 times more water would be required to grow nutritionally equivalent mature plants at an outdoor field-scale [109]. Furthermore, they also looked at time and found that they could grow this nutritionally equivalent amount of microgreens with less water in about 93–95% less time without Fertilizer, pesticides, or GHG emissions associated with food transport [109]. Therefore, this type of research is sorely needed, as it can inform production methods that can provide food-secure, highly dense, and nutritionally rich products.

In general, we showed that Fertilizer improved almost all of these RUE metrics. For a cost of only 48 cents per 30 L of Fertilized nutrient solution, which can be used for one layer for 8 days at a minimum, the biomass production gains are around a 60% increase in FW biomass and around a 25% increase in DW biomass. However, there are some associated difficulties with Fertilizer application, such as applying Nitric Acid, which must be safely and carefully handled as it is corrosive to human skin, and the labor associated with the daily application required to ensure the pH stays within the range necessary for nutrient utilization. Regarding SD, although per-plant production was higher at lower densities, there was a corresponding drop in SUE, indicating the advantages of using the High SD, especially in combination with Green and Far-Red containing Light Recipes that can benefit high-density environments. For instance, although the low seeding densities were more productive per plant, they required around 1.43 times the space compared to the High SD to produce equivalent biomass. This spatial inefficiency likely far outweighs any gains made by seed use efficiency that require the additional material inputs of adding additional growing systems. Although extensive RUE calculations are lacking for microgreens research, more thorough work has been carried out for other, more mature plants grown in similar controlled environments. A detailed review can be found in Orsini et al. [28], which discusses different tradeoffs and comparable production outcomes in different scenarios, from outdoor to greenhouse to PFAL results. This review, as well as our results here, show the direction that needs to be taken when considering the tradeoffs between production metrics and resource use efficiency metrics, helping to link research and practice to provide food-secure products.

## 5. Conclusions

This experiment showed improvements to microgreen morphology and biomass when using Far-Red light, especially at a ratio of Far-Red to PAR of around 0.30, which is above that found in natural sunlight (0.19). We therefore engineered a shade-avoidance response, which was accompanied by a high PFD of around 270 µmols/m^2^/s. We also found that the higher seeding densities had a greater response to the inclusion of Far-Red and Green light, which have lower absorption compared to Blue and Red wavelengths. However, it is important to consider that previous research has shown that the improvements offered by Far-Red light are dosage, wavelength, duration, and synergistically dependent. Furthermore, we found that the production of Far-Red photons was energetically less efficient than that of Red or Blue bands, due to the inefficiencies of the LED components and their method of being wired in series. The dynamic nature of light, known as the Emerson Enhancement Effect, and its capacity to induce changes in plants make it difficult to investigate causality in this type of research, but we have shown correlative relationships between different Light Recipes, including detailed analyses of their component wavelength bands and morphology and biomass outputs. Furthermore, although the engineered shade-avoidance High Far-Red Light Recipe had the significantly highest productivity metrics for both fresh weight and dry weight; there were corresponding increases in cost, energy, and light resource use inputs compared to the cheaper, lower intensity 24VHELEDs. Overall, we found that, when comparing LED types, the lower cost of 24-volt high-efficiency LEDs, which are cheap, lightweight, and require no programming, performed better in key resource use categories while also producing comparable biomass values. The advantages of the OSRAM lights would take several hundred generations to pay off due to their high upfront costs. In sum, we found that even with only four days of light treatments, there were significant responses of microgreens to different types of light, fertilization, and seeding densities. Unsurprisingly, we also found a significant and positive response of Fertilizer on microgreen morphology and biomass; seeding density also improved overall production, but this improvement was not linear, as there were per-plant production losses. We have therefore identified key production tradeoffs that can be useful for microgreen producers. Future experiments could benefit from expanding on this work by conducting purely orthogonal Far-Red studies from 0% up to greater than 40% and investigating the nutritional properties, sensory characteristics, and visual perception of microgreens.

## Figures and Tables

**Figure 1 plants-13-00124-f001:**
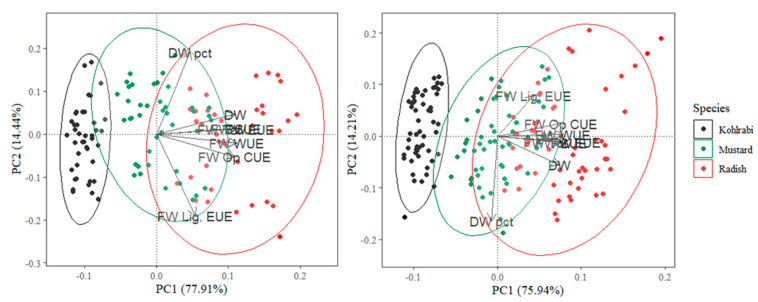
Biplot of principal component analysis (PCA) for FW (kg/m^2^), DW (kg/m^2^), FW Light EUE (g FW/kWh light), FW Total EUE (g FW/kWh total), FW Op CUE (g FW/dollar operating costs), FW WUE (g FW/L H_2_O), and FW SUE (g FW/m^2^/day) for Unfertilized (**left panel**) and Fertilized (**right panel**) treatments. Kohlrabi is shown in black, mustard in green, and radish in red.

**Figure 2 plants-13-00124-f002:**
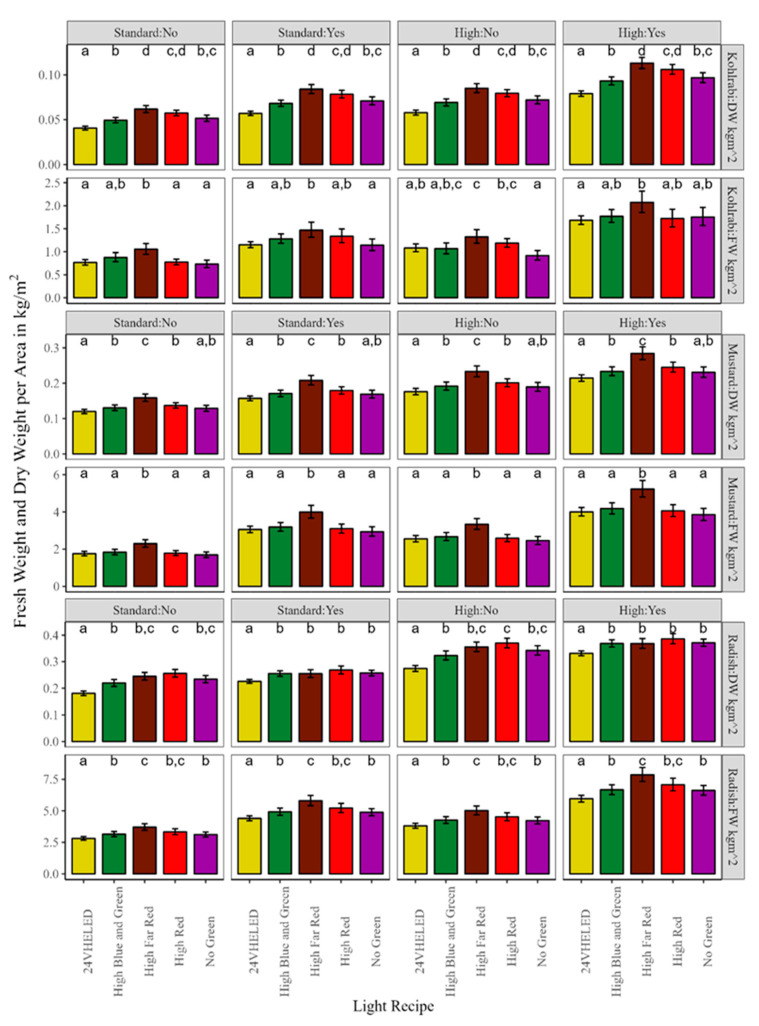
Facet graph detailing the kohlrabi (**top**), mustard (**middle**), and radish (**bottom**) FW and DW means (kg/m^2^) ± 95% confidence intervals at every combination of our experimental factors, seen as ‘Seeding Density: Fertilizer’. Coloured columns are associated with the Light Recipes: yellow (24VHELED), green (HBG), brown (HFR), red (HR), and purple (NG). Lowercase letters show the results of Tukey’s HSD within each panel, where common letters are not significantly different from one another at α = 0.05.

**Figure 3 plants-13-00124-f003:**
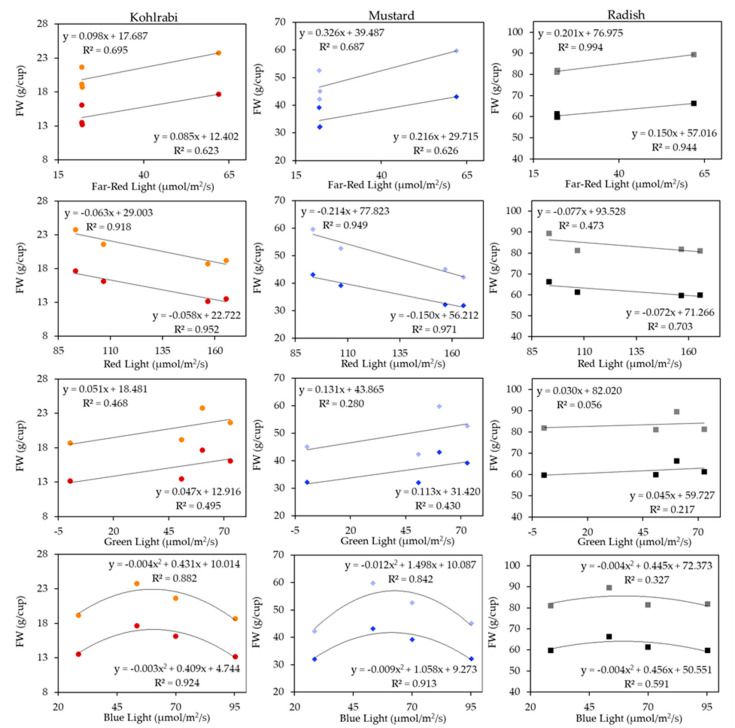
Regression plots between light components (**top** to **bottom**: Far-Red, Red, Green, and Blue) on the *x*-axis and FW (g/cup) on the *y*-axis for kohlrabi (red-hued circles), mustard (blue-hued diamonds), and radish (black-hued squares). The lighter hues in each corresponding figure indicate the High SD, while the darker colors indicate the Standard SD. Equations and R^2^ shown in each corresponding panel for the associated nearest SD.

**Figure 4 plants-13-00124-f004:**
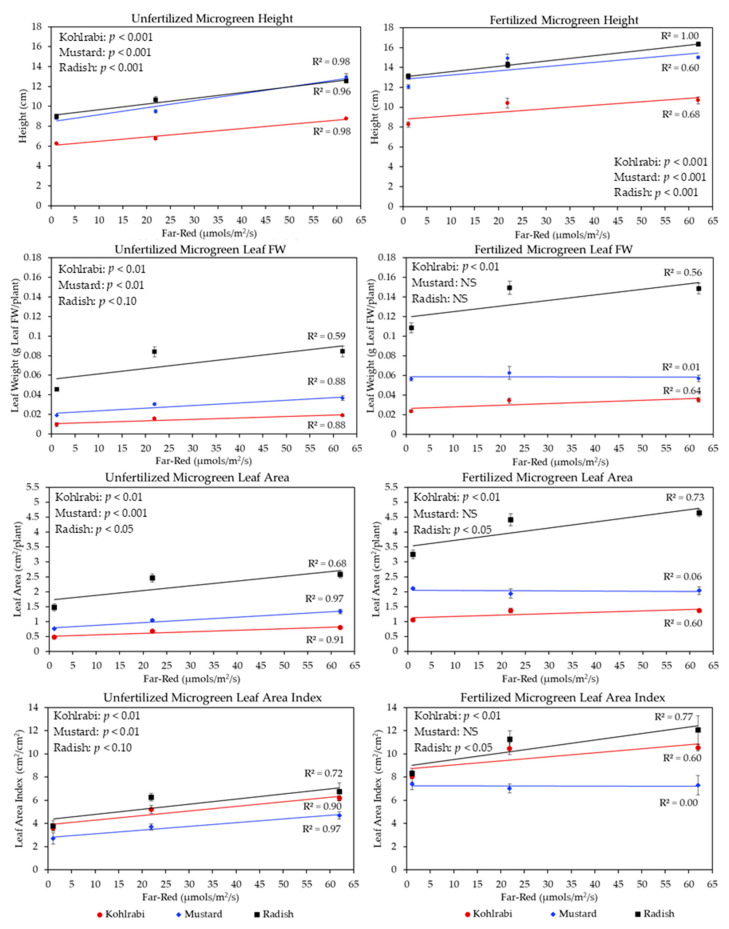
Regression plots between Far-Red light (µmols/m^2^/s) on the *x*-axis and height (cm; **top**), leaf Weight (g leaf FW/plant; **second from top**), leaf area (cm^2^/plant; **second from bottom**), and leaf area index (cm^2^/cm^2^; **bottom**) on the *y*-axis. Kohlrabi (red circles), mustard (blue diamond), and radish (black squares) values are shown as means ± SE, with corresponding R^2^. The *p*-values are from corresponding linear models.

**Figure 5 plants-13-00124-f005:**
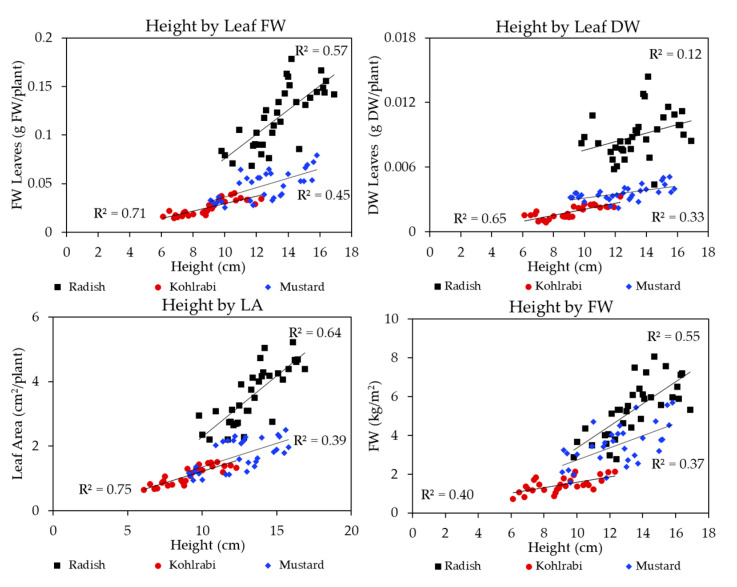
Regression plots between height (cm) on the *x*-axis and leaf FW (g leaf FW/plant), leaf DW (g leaf DW/plant), leaf area (cm^2^/plant), and total FW (kg/m^2^) on the *y*-axis. R^2^ and equations are shown next to each associated regression line.

**Figure 6 plants-13-00124-f006:**
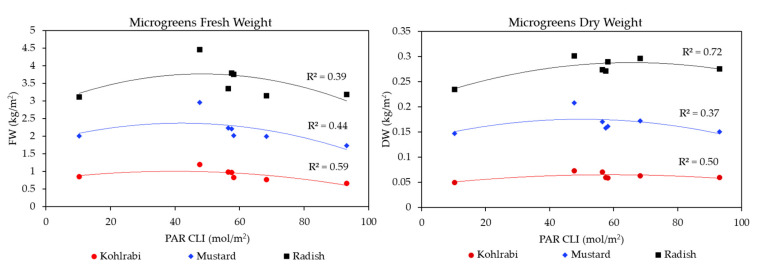
Scatterplots and quadratic curve fitting between FW (kg/m^2^) on the *y*-axis and Photosynthetically Active Radiation Cumulative Light Integral (PAR CLI; mol/m^2^) on the *x*-axis. Species shown as kohlrabi (red circle), mustard (blue diamond), and radish (black square) The corresponding R^2^ shown above each respective line.

**Figure 7 plants-13-00124-f007:**
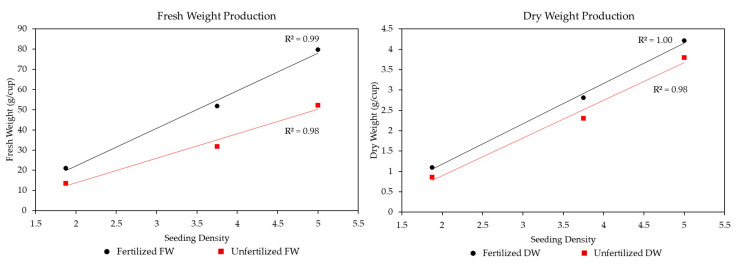
Regression plots showing the production of FW (**left panel**) and DW (**right panel**) biomass as a result of starting SD, in grams/cup. The red squares are treatments without Fertilizer, while the black circles received Fertilizer. Each point is the mean for that species at that treatment level. R^2^ shown for each corresponding line.

**Figure 8 plants-13-00124-f008:**
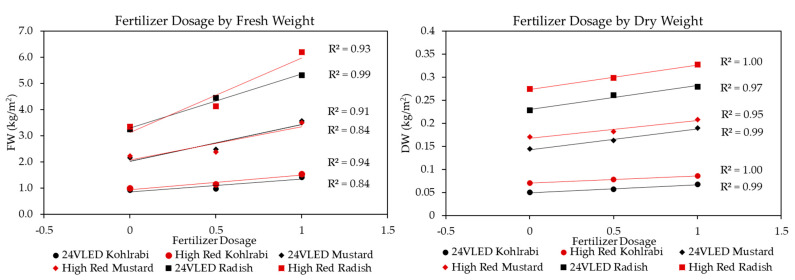
Regression plots showing the mean biomass of kohlrabi (circle), mustard (diamond), and radish (square) microgreens due to fertilization application at the levels of none (0), half duration (0.5), and full duration (1) for fresh weight (**left panel**) and dry weight (**right panel**) in units of kg/m^2^. Black lines show the 24VHELED, while the Red lines show the HR Light Recipe.

**Table 1 plants-13-00124-t001:** Spectrometric readings for experimental Light Recipe specific band light intensities (PFD; µmol/m^2^/s), PPFD (µmol/m^2^/s), PFD (µmol/m^2^/s), YPFD (µmol/m^2^/s), DLI (mol/m^2^), recipe component percentages (%), light band ratios, PPE, CIE, and Power Consumption (Watts). The Valoya reference Light Recipe is shown in the second column.

	Valoya Reference	High Red	No Green	High Blue Green	High Far-Red	24VHELED
Blue µmol/m^2^/s (400–500 nm)	x	28.64	95.38	70.04	53.43	7.91
Green µmol/m^2^/s (500–600 nm)	x	50.84	0.44	72.81	60.30	21.41
Red µmol/m^2^/s (600–700 nm)	x	165.50	156.60	106.75	93.30	15.68
Far-Red µmol/m^2^/s (700–800 nm)	x	21.93	22.06	21.88	61.96	1.10
PPFD µmol/m^2^/s (400–700 nm)	x	244.98	252.42	249.60	207.03	45.00
PFD µmol/m^2^/s (350–800 nm)	x	266.92	274.49	271.48	268.99	46.09
YPFD µmol/m^2^/s (350–800 nm)	x	219.90	222.34	212.36	186.07	39.75
DLI (mol/m^2^)	x	15.37	15.81	15.64	15.49	2.65
% Blue (400–500 nm)	10%	10.73%	34.75%	25.80%	19.86%	17.16%
% Green (500–600 nm)	19%	19.05%	0.16%	26.82%	22.42%	46.45%
% Red (600–700 nm)	63%	62.01%	57.05%	39.32%	34.69%	34.02%
% Far-Red (700–800 nm)	8%	8.22%	8.04%	8.06%	23.03%	2.38%
B:R	0.16	0.17	0.61	0.66	0.57	0.50
B:G	0.53	0.56	216.77	0.96	0.89	0.37
G:R	0.30	0.31	0.00	0.68	0.65	1.37
R:FR	7.88	7.55	7.10	4.88	1.51	14.25
FR:PAR	0.09	0.09	0.09	0.09	0.30	0.02
PPE	x	0.79	0.84	0.83	0.75	0.86
CIE	60.00	22.01	0.00	60.78	60.04	81.29
Power Consumption (Watts)	x	171	141	192	189	48

**Table 2 plants-13-00124-t002:** Means ± standard deviation for Low Zone Temperature (Temp.; °C), Low Zone Relative Humidity (RH; %), Low Zone Vapor Pressure Deficit (VPD; kPa), High Zone Temperature (Temp.; °C), High Zone RH (%), High Zone VPD (kPa), and Climate Chamber Temp (CC1; °C). Values are shown for the first experimental period (Period One) in the first row and the second experimental period (Period Two) in the second row. ND = no data.

Period	Low Zone Temp. (°C)	Low Zone RH (%)	Low Zone VPD (kPa)	High Zone Temp. (°C)	High Zone RH (%)	High Zone VPD (kPa)	CC Temp (°C)
One	21.6 ± 0.2	69.6 ± 4.5	0.79 ± 0.12	21.8 ± 0.2	68.8 ± 3.6	0.81 ± 0.10	21.1 ± 0.1
Two	21.5 ± 0.2	ND	ND	21.9 ± 0.6	66.1 ± 5.1	0.89 ± 0.15	21.0 ± 0.1

**Table 3 plants-13-00124-t003:** Means ± standard deviation for pH and Electrical Conductivity (EC; mS/cm) for Period One and Period Two, with and without Fertilizer (Fert. and Unfert., respectively).

Measurement	Period One Unfert.	Period Two Unfert.	Period One Fert.	Period Two Fert.
pH	6.57 ± 0.55	6.56 ± 0.28	6.20 ± 0.20	6.18 ± 0.19
EC	0.81 ± 0.04	0.86 ± 0.06	1.86 ± 0.09	1.81 ± 0.08

**Table 4 plants-13-00124-t004:** Type I ANOVA table results for interspecies analysis. Eta-squared (η^2^) values and *p*-values are shown for FW and DW.

Source of Variation	df	FW (η^2^)	FW (p)	DW (η^2^)	DW (p)
Species	2	65.28%	<0.001	77.24%	<0.001
Light Recipe	4	2.09%	<0.001	2.32%	<0.001
Fertilizer	1	15.65%	<0.001	3.78%	<0.001
Seeding Density	1	6.81%	<0.001	10.28%	<0.001
Light Recipe:Fertilizer	4	0.02%	0.98	0.09%	0.42
Fertilizer:Seeding Density	1	0.21%	0.02	0.04%	0.17
Light Recipe:Seeding Density	4	0.06%	0.83	0.10%	0.39
Light Recipe:Fertilizer:Seeding Density	4	0.01%	1.00	0.03%	0.83
Error		9.88%		6.11%	

**Table 5 plants-13-00124-t005:** Means ± SE for kohlrabi, mustard, and radish specific leaf area (SLA; cm^2^/g leaf DW) with Fertilizer (Fert.) and without Fertilizer (Unfert.).

	Kohlrabi (Unfert.)	Kohlrabi (Fert.)	Mustard (Unfert.)	Mustard (Fert.)	Radish (Unfert.)	Radish (Fert.)
24VHELED	511.61 ± 10.66	814.45 ± 25.5	336.96 ± 50.04	601.42 ± 68.57	440.39 ± 131.75	512.53 ± 23.53
Low Far-Red OSRAM	400.25 ± 34.19	583.03 ± 21.00	309.73 ± 17.64	408.33 ± 84.12	282.98 ± 8.13	398.82 ± 15.83
High Far-Red OSRAM	557.73 ± 20.47	559.67 ± 14.57	415.7 ± 23.56	473.77 ± 7.96	352.33 ± 13.07	465.73 ± 9.97

**Table 6 plants-13-00124-t006:** Table of mean DW Percent (g DW/g FW) for kohlrabi, mustard, and radish microgreens for Light Recipe, with and without fertilization (Yes and No, respectively). Means are averaged across SDs.

Species	Fertilizer	24VHELED	High Blue and Green	High Far-Red	High Red	No Green
Kohlrabi	No	5.25%	6.09%	6.15%	7.14%	7.10%
Kohlrabi	Yes	4.84%	5.29%	5.57%	5.64%	6.15%
Mustard	No	6.48%	7.21%	6.86%	7.71%	7.94%
Mustard	Yes	5.26%	5.46%	5.06%	5.95%	5.67%
Radish	No	6.52%	7.14%	6.66%	7.85%	7.71%
Radish	Yes	5.17%	5.42%	4.57%	5.29%	5.45%

**Table 7 plants-13-00124-t007:** Fertilizer application improvements for kohlrabi, mustard, and radish (FW and DW) Values shown as the mean ratio (grams of Fertilized biomass/grams of Unfertilized biomass) for each Light Recipe. The total average is shown at the bottom as mean ± SE.

Light Recipe	Kohlrabi (FW)	Kohlrabi (DW)	Mustard (FW)	Mustard (DW)	Radish (FW)	Radish (DW)
24VHELED	1.51	1.35	1.67	1.32	1.64	1.22
High Blue and Green	1.57	1.36	1.75	1.31	1.53	1.16
High Far-Red	1.48	1.36	1.49	1.13	1.50	1.04
High Red	1.58	1.26	1.61	1.25	1.59	1.05
No Green	1.74	1.51	1.76	1.26	1.55	1.09
Total Average	1.58 ± 0.05	1.36 ± 0.04	1.66 ± 0.05	1.25 ± 0.03	1.56 ± 0.03	1.11 ± 0.02

**Table 8 plants-13-00124-t008:** Microgreen Unfertilized and Fertilized FW mean ratio ± SE (High SD FW biomass in grams/Standard SD FW biomass in grams).

Treatment	Kohlrabi	Mustard	Radish
Unfertilized FW	1.33 ± 0.06	1.45 ± 0.01	1.38 ± 0.04
Fertilized FW	1.42 ± 0.04	1.31 ± 0.04	1.35 ± 0.02
**Average**	**1.37 ± 0.05**	**1.38 ± 0.02**	**1.36 ± 0.03**

**Table 9 plants-13-00124-t009:** Average Resource Use Efficiencies across species ± SE for Light Use Efficiency (LUE), Light Energy Use Efficiency (EUE), Total EUE, Water Use Efficiency (WUE), Surface Use Efficiency (SUE), and Cost Use Efficiency (CUE) of operating costs Lowercase letters show the results of Tukey’s HSD, where common letters are not significantly different from one another at α = 0.05.

Fert.	Light Recipe	LUE (g FW/mol PFD)	Light EUE (g FW/kWh)	Total EUE (g FW/kWh)	WUE (g FW/L H_2_O)	SUE (g FW/m^2^)	CUE (g FW/Dollar)
No	24VHELED	45.75 ± 3.83 c	1116.04 ± 93.44 c	2.27 ± 0.19 a	57.14 ± 4.79 b,c	263.59 ± 22.07 a	70.84 ± 5.36 c
No	HBG	8.55 ± 1.10 a	189.00 ± 24.23 a	2.48 ± 0.32 b	55.25 ± 7.08 a,b	290.30 ± 37.21 b	54.33 ± 6.49 a
No	HFR	10.67 ± 1.28 b	237.25 ± 28.52 b	3.06 ± 0.37 c	63.13 ± 7.59 c	358.73 ± 43.11 c	67.60 ± 7.58 b
No	HR	7.74 ± 0.80 a	188.85 ± 19.62 a,b	2.21 ± 0.23 b	46.84 ± 4.87 a	258.35 ± 26.84 b	50.77 ± 4.96 a
No	NG	8.01 ± 1.16 a	243.78 ± 35.14 b	2.35 ± 0.34 a,b	50.57 ± 7.29 a	274.98 ± 39.64 a,b	56.65 ± 7.57 a
Yes	24VHELED	74.51 ± 4.59 b	1817.46 ± 111.98 c	3.70 ± 0.23 a	93.05 ± 5.73 b,c	429.26 ± 26.45 a	114.67 ± 6.48 c
Yes	HBG	15.77 ± 1.87 a	301.93 ± 26.66 a	3.96 ± 0.35 b	88.26 ± 7.79 a,b	463.77 ± 40.94 b	86.56 ± 7.22 a
Yes	HFR	15.95 ± 1.94 a	354.62 ± 43.12 b	4.58 ± 0.56 c	94.36 ± 11.47 c	536.19 ± 65.19 c	100.47 ± 11.42 b
Yes	HR	14.03 ± 1.84 a	342.30 ± 44.87 a,b	4.00 ± 0.53 b	84.90 ± 11.13 a	468.26 ± 61.38 b	90.72 ± 11.06 a
Yes	NG	14.95 ± 1.45 a	454.71 ± 44.13 b	4.39 ± 0.43 a,b	94.33 ± 9.15 a	512.91 ± 49.78 a,b	104.46 ± 9.42 a

## Data Availability

The data presented in this study are available on request from the corresponding author. The data are not currently publicly available due to its involvement in ongoing project activities.

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
