# Peer review of "The Effects of Light Spectrum and Intensity, Seeding Density, and Fertilization on Biomass, Morphology, and Resource Use Efficiency in Three Species of Brassicaceae Microgreens"

_plants, 2024, doi:10.3390/plants13010124_

Round 1

Reviewer 1 Report

Comments and Suggestions for Authors

There are some problems in the manuscript listed as follow:

1 In the 2. Materials and Methods, some figures or tables should be moved to the supplements.

2 the writing of the Result should be refined and deleted the redundant explanation.

Comments on the Quality of English Language

The writing of this manu should be improved for the refinement.

Author Response

Reviewer 1 Comments:

Query: There are some problems in the manuscript listed as follow:

1 In the 2. Materials and Methods, some figures or tables should be moved to the supplements.

Response: Thank you for your comments. I have moved two figures (photon efficiency and visualized light recipes) and one table (seed sowing densities, germination, etc.) from the Materials and Methods to the Supplementary Materials. See lines 199, 222, and 302.

Query: 2 the writing of the Result should be refined and deleted the redundant explanation.

Response: Thank you for your comments. We have incorporated your suggestion by reducing the systematic results descriptions and tried to put them into the same sentences for easier reading, as well as making additional quality changes to reduce redundant explanation. See track changes for all changes in the results section,  but also lines 645-47, 711-713, 743-46, 775, 786, 1115-116, 1137, and 1509 for specific examples.  

Additionally, we have varied the phrasing and language of each species-specific analysis section in the supplementary materials along the same lines.

Query: The writing of this manu should be improved for the refinement.

Response: We believe this is in keeping with the previous query and have taken it in consideration when making our revisions of results and discussion sections.

Reviewer 2 Report

Comments and Suggestions for Authors

Light has a significant role in the process of plant growth. The major signal sensed by plants is represented by light, and substantial research has shown that numerous characteristics of light, such as its quality, intensity, and duration, have significant regulatory impacts on the morphogenesis, physiological metabolism, growth and development, and nutritional quality of plants. The primary constraints impeding the development of microgreen technologies in plant factories are the substantial investment costs and the energy requirements for artificial light. To achieve multi-level plant production, it is necessary to utilize light sources that have high energy conversion rate to the specific light wavelengths utilized by plants during photosynthesis, while minimizing heat generation. This study provides valuable information on the impact of light quality, fertilization, and seed density on the morphological changes, biomass accumulation, and resource input cost of microgreens. The manuscript has a comprehensive scope and is written satisfactorily. The "Materials and Methods" part provides comprehensive and detailed information, while rigorous statistical analysis confirms the findings presented in the "Results" section. The manuscript has remarkable written proficiency and provides a thorough and thorough evaluation. Overall, the main conclusions and concepts are adequate and well-justified. The manuscript demonstrates the capacity to be suitable for publication.

Author Response

Reviewer 2 Comments:

Query: Light has a significant role in the process of plant growth. The major signal sensed by plants is represented by light, and substantial research has shown that numerous characteristics of light, such as its quality, intensity, and duration, have significant regulatory impacts on the morphogenesis, physiological metabolism, growth and development, and nutritional quality of plants. The primary constraints impeding the development of microgreen technologies in plant factories are the substantial investment costs and the energy requirements for artificial light. To achieve multi-level plant production, it is necessary to utilize light sources that have high energy conversion rate to the specific light wavelengths utilized by plants during photosynthesis, while minimizing heat generation. This study provides valuable information on the impact of light quality, fertilization, and seed density on the morphological changes, biomass accumulation, and resource input cost of microgreens. The manuscript has a comprehensive scope and is written satisfactorily. The "Materials and Methods" part provides comprehensive and detailed information, while rigorous statistical analysis confirms the findings presented in the "Results" section. The manuscript has remarkable written proficiency and provides a thorough and thorough evaluation. Overall, the main conclusions and concepts are adequate and well-justified. The manuscript demonstrates the capacity to be suitable for publication.

Response: Thank you for your response. We are glad to hear that the manuscript is suitable for publication.

Reviewer 3 Report

Comments and Suggestions for Authors

Paper is very well written. It presents useful resoults of effects of light spectrum on biomass and morphology of kohlrabi, mustard and radish.

Although there are not presented very important quality data, such as glucosinolanes, phenolics, vitamins content, but results in this paper are enough to publish as a preliminary study.

Some minor issues I found:

a) add source (provider) of seeds that were used in experiment;

b) name of plants not with capital letter e.g. L. 166; The same with "Leaf Area", "Heright" on Fig. 7 and in othr places of manuscript;

c) the same for "standard error";

d) please present only one sig. digits for temperature measurements in Table 3  In real it is impossible to control temp. with 0.01 oC accuracy;

e) add info, why authors used a vapor deficit in experiment.

Author Response

Reviewer 3 Comments:

Query: Paper is very well written. It presents useful resoults of effects of light spectrum on biomass and morphology of kohlrabi, mustard and radish.

Although there are not presented very important quality data, such as glucosinolanes, phenolics, vitamins content, but results in this paper are enough to publish as a preliminary study.

Response: Thank you for your comments. We also agree, and we would have liked to conduct assays on Glucosinolates, vitamins, or minerals.

Query: Some minor issues I found: a) add source (provider) of seeds that were used in experiment;

Response: Done. See line 195-96.

Query: b) name of plants not with capital letter e.g. L. 166; The same with "Leaf Area", "Heright" on Fig. 7 and in othr places of manuscript;

Response: We have incorporated your comment in the manuscript. Our convention was to capitalize specific linear model or ANOVA effects. However, we have systematically changed that convention to reflect your comment for the plant names. Additionally, we agree that capitalizing Height, Leaf Area, or other measurements is unnecessary and have changed this accordingly. See lines 136, 197,  191-93, 547, 788, 800, 1030, etc., for examples, but we have made a systematic and thorough change throughout the entire manuscript for consistency.

Query: c) the same for "standard error";

Response: We agree, and have incorporated that change. See lines 199-200.

Query: d) please present only one sig. digits for temperature measurements in Table 3  In real it is impossible to control temp. with 0.01 oC accuracy;

Response: We agree, and have changed the number of significant digits for temperature. See line 329.

Query: e) add info, why authors used a vapor deficit in experiment.

Response: Vapor pressure deficit was calculated to integrate the combination of our temperature and relative humidity conditions and provide information to those who use that unit about what type of atmospheric conditions we experimented on our Microgreens in. We used it with the aim of achieving a targeted VPD of around 0.85. See line 321-24 for my explanation.